# A single power stroke by ATP binding drives substrate translocation in a heterodimeric ABC transporter

**Erich Stefan, Susanne Hofmann, Robert Tampé***

Institute of Biochemistry, Goethe University Frankfurt, Biocenter, Germany

**Abstract** ATP-binding cassette (ABC) transporters constitute the largest family of primary active transporters, responsible for many physiological processes and human maladies. However, the mechanism how chemical energy of ATP facilitates translocation of chemically diverse compounds across membranes is poorly understood. Here, we advance the quantitative mechanistic understanding of the heterodimeric ABC transporter TmrAB, a functional homolog of the transporter associated with antigen processing (TAP) by single-turnover analyses at single-liposome resolution. We reveal that a single conformational switch by ATP binding drives unidirectional substrate translocation. After this power stroke, ATP hydrolysis and phosphate release launch the return to the resting state, which facilitates nucleotide exchange and a new round of substrate binding and translocation. In contrast to hitherto existing steady-state assays, our single-turnover approach uncovers the power stroke in substrate translocation and the tight chemomechanical coupling in these molecular machines.

**\*For correspondence:**
tampe@em.uni-frankfurt.de

**Competing interests:** The authors declare that no competing interests exist.

## Introduction

Ubiquitous in all phyla of life, ABC transporters shuttle chemically diverse compounds across cell membranes, ranging from peptides and proteins to lipids, metabolites, antibiotics, and drugs (*Dean et al., 2001*; *Rees et al., 2009*; *Thomas and Tampé, 2018*). Substrate transport is facilitated by large conformational changes, which are chemomechanically coupled to the hydrolysis of ATP (*Abele and Tampé, 2004*; *Higgins and Linton, 2004*; *Senior and Gadsby, 1997*). ABC exporters are composed of two nucleotide-binding domains (NBDs), responsible for ATP binding and hydrolysis, and two transmembrane domains (TMDs) forming the substrate translocation pathway. ABC exporters are involved in multifold physiological pathways and human diseases, ranging from cystic fibrosis to acquired multidrug resistance in cancer chemotherapy (*Csanády et al., 2019*; *Robey et al., 2018*). The transporter associated with antigen processing (TAP) plays a critical role in the adaptive immune response against intracellular pathogens and tumors (*Blum et al., 2013*; *Trowitzsch and Tampé, 2020*). Notably, more than half of all human ABC transporters have two functionally asymmetric ATP-binding sites (canonical *vs.* non-canonical). The *Thermus thermophilus* multidrug resistance transport complex TmrAB represents a structural and functional homolog of the TAP complex with an overlapping peptide specificity and can restore antigen presentation in human TAP-deficient cells (*Kim et al., 2015*; *Nöll et al., 2017*; *Zutz et al., 2011*). In recent years, the molecular understanding of the conformational space of ABC transporters has been expanded. In particular, eight high-resolution cryo-EM structures of TmrAB in different states and conformations were determined, comprising two inward-facing (IF), four outward-facing (OF), and two asymmetric post-hydrolysis states (*Hofmann et al., 2019*). Based on this detailed insight, two major transitions from IF-to-OF and OF-to-IF conformation can be defined. However, the functional roles of ATP binding and ATP hydrolysis in the substrate translocation cycle are poorly defined. Owing to the fast ATP

turnover in ABC transporters paired with the hydrophobicity of most substrates, it has been intrinsically difficult to tackle these indispensable questions.

Notably, exporters to our knowledge ABC have not been analyzed by single-turnover assays, which would require full transport activity of all correctly reconstituted ABC transporters in liposomes paired with sensitive detection of the translocated substrate. Thus, it remained largely elusive until now whether a single-substrate translocation event is mediated by ATP binding or ATP hydrolysis and how substrate and nucleotide binding affect the major conformational changes. Furthermore, some ABC exporters have been described as uncoupled molecular machines that undergo multiple rounds of ATP hydrolysis per translocated substrate (*Bock et al., 2019*; *Eytan et al., 1996*; *Nöll et al., 2017*; *Zehnpfennig et al., 2009*; *Zollmann et al., 2015*). The significance of these futile cycles, however, is largely unclear.

Here, we address these questions by establishing single-turnover assays. The translocation complex TmrAB was reconstituted in lipid nanodiscs or liposomes providing an optimal lipid environment (*Bechara et al., 2015*) for full activity and conformational freedom (*Hofmann et al., 2019*). We delineate that a single conformational switch from the IF-to-OF conformation, triggered by ATP binding, drives unidirectional substrate translocation across the membrane. After ATP hydrolysis in and phosphate release from the canonical site, the OF-to-IF transition is launched, which allows for nucleotide exchange and a new round of substrate processing. Our single-turnover analyses provide quantitative insights into the inner mechanics and coupling of ATP binding, hydrolysis, and substrate translocation of ABC transporters.

## Results

### The IF-to-OF switch induced by ATP binding precludes substrate binding

ABC transporters pass through very transient pre- and post-hydrolysis states. In order to kinetically separate the events of ATP binding and hydrolysis, we substituted the catalytic glutamate by glutamine at the canonical ATP site (TmrA$^{E523Q}$B, TmrA$^{EQ}$B) to slowdown the ATP turnover. The transporter was reconstituted in lipid nanodiscs (*McLean et al., 2018*), maintaining its optimal membrane environment for function and conformational dynamics (*Hofmann et al., 2019*). By scintillation proximity assays (SPA), we confirmed that wildtype TmrAB and TmrA$^{EQ}$B are equivalent in equilibrium peptide- and nucleotide-binding (*Figure 1—figure supplement 1*). We therefore utilized TmrA$^{EQ}$B with a slowdown in ATP hydrolysis for single-turnover studies to probe the roles of ATP binding, ATP hydrolysis, and both subsequent conformational transitions (IF-to-OF and return) for substrate translocation. When comparing our high-resolution cryo-EM structures of IF and OF conformers (*Hofmann et al., 2019*), we found that key residues involved in peptide binding are rearranged upon the IF-to-OF transition. Transmembrane helices TM2, TM3, and TM4, framing the substrate-binding site in the IF conformation, get closely packed together in the OF state to form a tightly sealed intracellular gate (*Figure 1a*). To monitor this conformational switch, we analyzed the equilibrium peptide binding to IF and OF TmrA$^{EQ}$B. After the IF-to-OF transition induced by ATP binding, the association of radiolabeled peptides to TmrA$^{EQ}$B was precluded (*Figure 1b*). The impairment of substrate binding was also shown by fluorescence anisotropy using the fluorescent C4F peptide (*Figure 1c*). Loss of peptide binding correlates with the concentration of ATP added to induce the IF-to-OF transition, which is monitored by fluorescence anisotropy using fluorescent peptides (*Figure 1d*) and by SPA utilizing radiolabeled peptides (*Figure 1—figure supplement 2*). The EC$_{50}$ of 30 µM is consistent with the equilibrium dissociation constant of Mg$^{2+}$-ATP (*Figure 1e*). It is worth mentioning that we observed a similar loss of substrate binding after nucleotide trapping in the OF conformation by ATPγS, a slowly hydrolyzing ATP analog (*Figure 1—figure supplement 2*). Using the same experimental conditions, binding of radiolabeled and fluorescent peptides was not affected by Mg$^{2+}$-ADP (*Figure 1d* and *Figure 1—figure supplement 2*), although Mg$^{2+}$-ADP and Mg$^{2+}$-ATP display similar affinities for TmrAB in the IF conformation (*Figure 1e*). In conclusion, only ATP and not ADP can drive the IF-to-OF transition, which in turn abrogates high-affinity substrate binding to the ABC transporter.

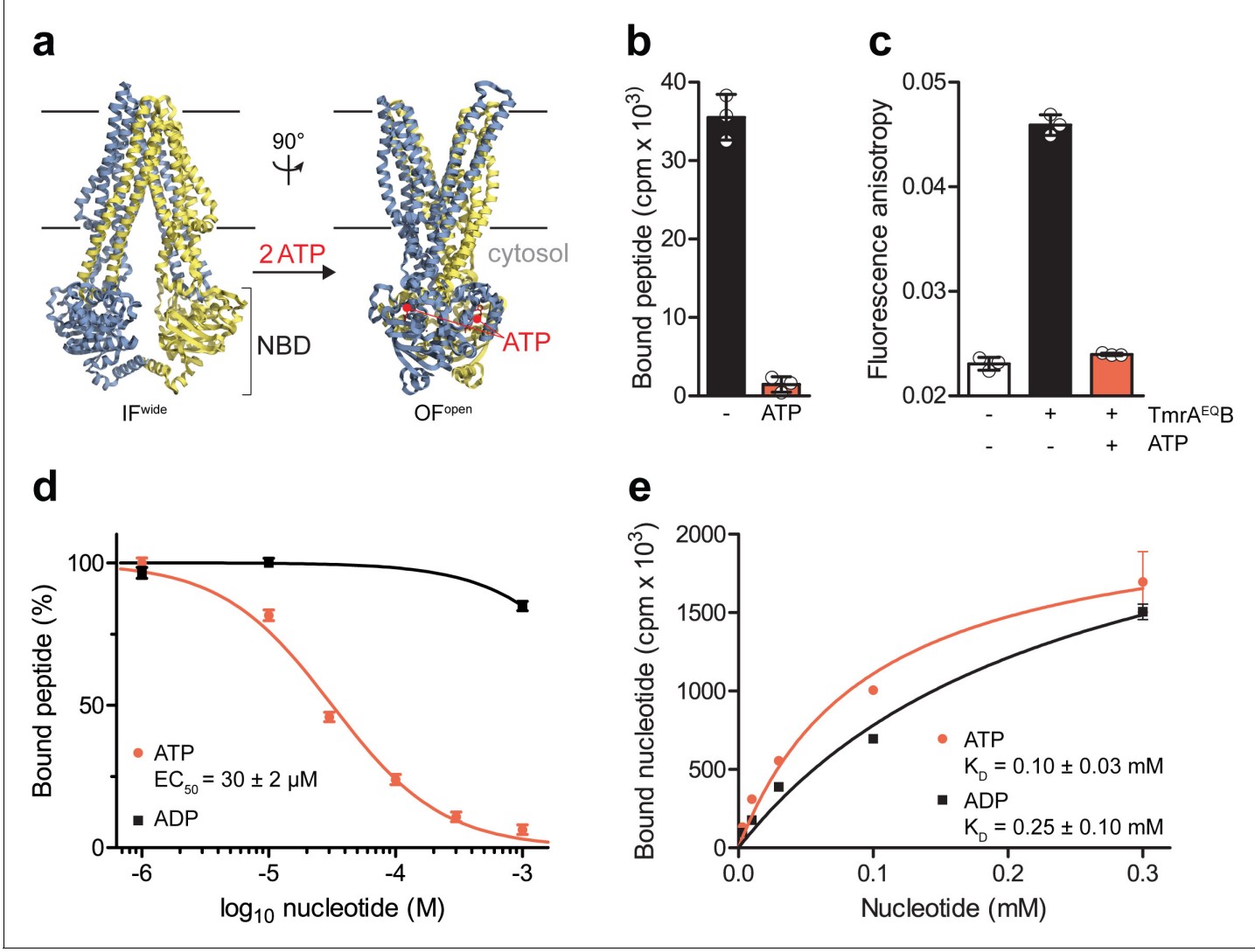

**Figure 1.** The IF-to-OF transition by ATP binding precludes substrate binding. (**a**) Scheme of the IF-to-OF transition by comparing the IF$^{wide}$ and OF$^{open}$ conformation of TmrAB. TmrA, TmrB, and ATP are shown in cyan, yellow, and red, respectively. (**b**) Loss in peptide binding after the ATP-induced IF-to-OF transition. TmrA$^{EQ}$B (0.2 μM) in Nds was incubated for 5 min at 45 ˚C with the R9LQK peptide (10 μM, supplemented with $^3$H-R9L) in the presence and absence of Mg$^{2+}$-ATP (1 mM). Peptide binding was monitored by SPA. Signals were background-corrected and multiplied by the dilution factor of radiolabeled peptides. (**c**) Peptide binding after the IF-to-OF switch. The fluorescence anisotropy of C4F peptide (50 nM) was determined at $\lambda_{ex/em}$ = 485/520 nm in the absence and presence of TmrA$^{EQ}$B in Nds (1 μM). To distinguish between the IF and OF conformation, TmrA$^{EQ}$B was incubated for 5 min at 45 ˚C with and without Mg$^{2+}$-ATP (1 mM). (**d**) Nucleotide-dependent peptide binding. TmrA$^{EQ}$B in Nds (1 μM) was incubated for 5 min at 45 ˚C with increasing concentrations of nucleotide. (**e**) Equilibrium binding affinity of ATP or ADP analyzed by scintillation proximity assay. TmrA$^{EQ}$B (0.2 μM) was incubated with increasing concentration of ATP or ADP supplemented with $^3$H-ATP or $^3$H-ADP, respectively. The mean of three independent experiments was analyzed by a one-site binding model, $K_{D, ATP}$ = 0.10 ± 0.03 mM, $K_{D, ADP}$ = 0.25 ± 0.10 mM. In (**b–e**) the mean ± SD (n = 3) is displayed.

The online version of this article includes the following source data and figure supplement(s) for figure 1:

**Source data 1.** Source Data to *Figure 1b-e* and *Figure 1—figure supplement 1* and *2*.
**Figure supplement 1.** Peptide binding to TmrAB reconstituted in lipid nanodiscs.
**Figure supplement 2.** Nucleotide-dependent peptide binding of TmrA$^{EQ}$B in lipid nanodiscs.

## A single ATP turnover and phosphate release trigger the OF-to-IF return

After converting the TmrA$^{EQ}$B complexes into the OF conformation (5 min, 45 ˚C) and removal of free ATP by rapid gel filtration, we analyzed the occlusion of either [α$^{32}$P]-ATP or [γ$^{32}$P]-ATP in

TmrA$^{EQ}$B by thin layer chromatography and autoradiography. We then monitored the OF-to-IF transition by (i) the release of the occluded radiolabeled nucleotides, (ii) the restoration of peptide binding, as well as (iii) the ATP hydrolysis and phosphate release (*Figure 2a,b*). By scintillation proximity assays, we observed a mono-exponential decay of the nucleotides occluded in TmrA$^{EQ}$B and an underlying decline of binding signal over time typical of SPA. The half-life of the nucleotide-occluded state at 20 °C was 24 min, which is in accordance with the ATP turnover of 29 ± 1 min at room

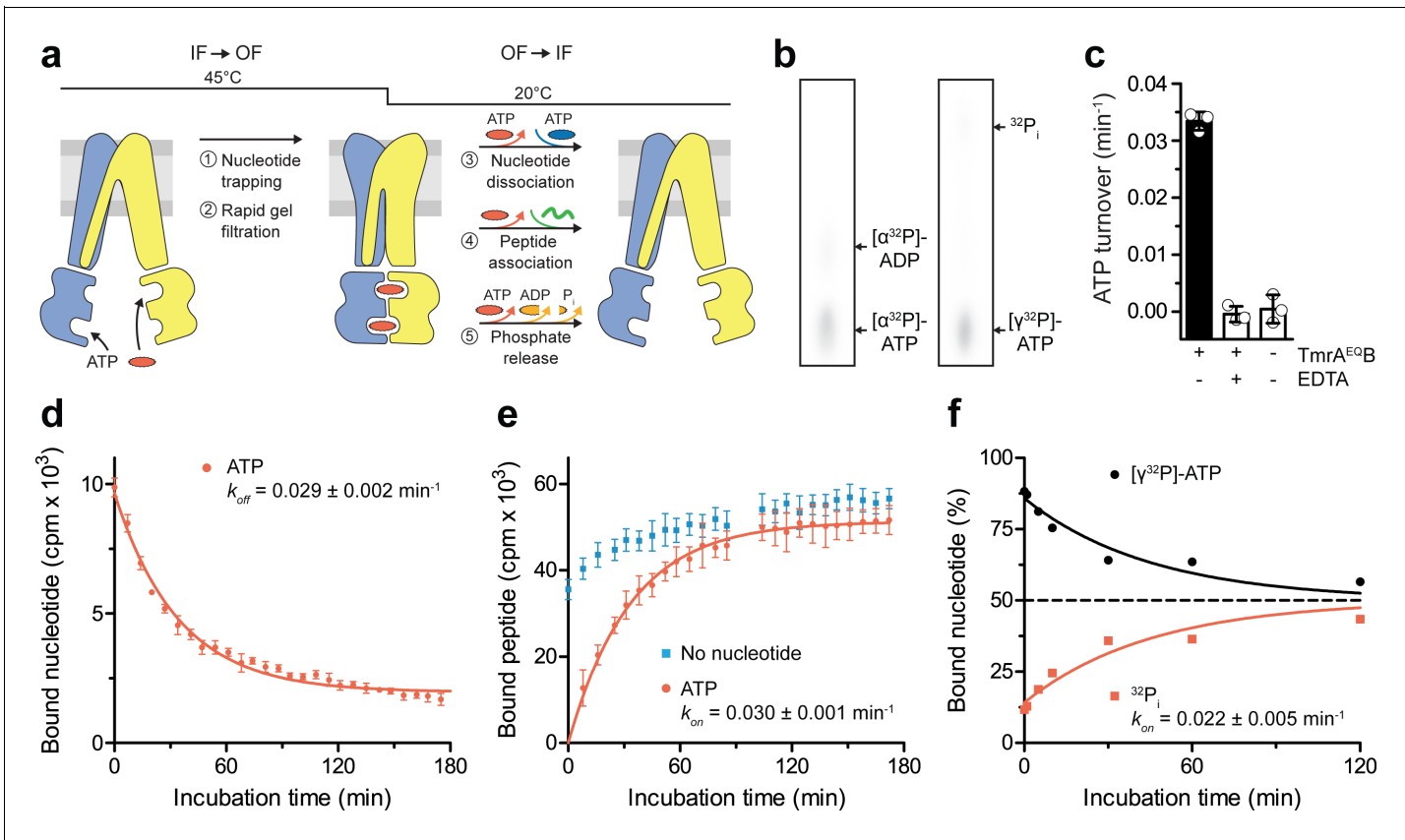

**Figure 2.** A single ATP turnover and phosphate release launches the OF-to-IF return. (**a**) Experimental scheme monitoring the OF-to-IF transition. (**b**) Nucleotide occlusion after the IF-to-OF transition. TmrA$^{EQ}$B in Nds (2 μM) was incubated for 5 min at 45 °C with Mg$^{2+}$-ATP (1 mM, traced with [α$^{32}$P]-ATP or [γ$^{32}$P]-ATP). ATP (2 mM) was added and unbound nucleotides were removed by rapid gel filtration. The radiolabeled nucleotides occluded by TmrA$^{EQ}$B were identified by thin layer chromatography and autoradiography. Representative autoradiograms for three independent triplicates are shown. (**c**) ATPase activity. TmrA$^{EQ}$B (1 μM) was incubated with Mg$^{2+}$-ATP (1 mM, traced with [γ$^{32}$P]-ATP) for 5 min at 45 °C and 15 min at 20 °C. Control reactions were performed in the presence of EDTA (10 mM). ATP autohydrolysis was recorded in the absence of TmrA$^{EQ}$B. The release of [γ$^{32}$P] was quantified by thin layer chromatography and autoradiography. Data were normalized to autohydrolysis and the mean values ± SD (n = 3) are displayed. (**d**) Nucleotide dissociation upon the OF-to-IF return. TmrA$^{EQ}$B in Nds (0.2 μM) was incubated with Mg$^{2+}$-ATP (3 μM, traced with $^3$H-ATP) for 5 min at 45 °C. Unbound nucleotides were removed by rapid gel filtration. An excess of Mg$^{2+}$-ATP (1 mM) was added to prevent any reassociation of released nucleotides. Nucleotide dissociation was followed by SPA at 20 °C. The mean ± SD (n = 3) is displayed and monoexponentially fitted, leading to a dissociation rate constant $k_{off}$ of 0.029 ± 0.002 min$^{-1}$ ($\tau_{1/2}$ = 21–26 min 95% confidence interval). (**e**) Peptide rebinding along the OF-to-IF return. TmrA$^{EQ}$B (0.2 μM) complexes were converted to the OF state as described in **a**). The excess of ATP was removed by rapid gel filtration. R9LQK peptide binding (10 μM, traced with $^3$H-R9L) was monitored by SPA at 20 °C. The mean ± SD (n = 3) is displayed and monoexponentially fitted, leading to an apparent association rate $k_{on}$ of 0.030 ± 0.001 min$^{-1}$ ($\tau_{1/2}$ = 21–25 min 95% confidence interval). (**f**) Release of inorganic phosphate along the OF-to-IF return. TmrA$^{EQ}$B in Nds (2 μM) was incubated with Mg$^{2+}$-ATP (1 mM, traced with [γ$^{32}$P]-ATP) and converted to the OF state as described in **b**). ATP (2 mM) was added and unbound nucleotides were removed by rapid gel filtration. Subsequently, a large excess of unlabeled ATP (2 mM) was added to prevent reassociation of [γ$^{32}$P]-ATP and thus to guarantee a single-turnover round of ATP hydrolysis. Nucleotides and released phosphate were analyzed as in panel **c**). It is important to note that some ATP is hydrolyzed before the TLC analysis. The mean ± SD (n = 3) is displayed and monoexponentially fitted, leading to inversely correlated rates of ATP turnover and phosphate release ($k_{on}$ ($^{32}$P$_i$)=$k_{off}$ ([γ$^{32}$P]-ATP)=0.022 ± 0.005 min$^{-1}$). The online version of this article includes the following source data and figure supplement(s) for figure 2:

**Source data 1.** Source Data to *Figure 2c-f* and *Figure 2—figure supplement 1*.
**Figure supplement 1.** Dissociation of occluded ATP.

temperature (*Figure 2c,d*). The half-life of the nucleotide-occluded state was slightly affected by temperature. At 4 ˚C, the half-life was prolonged to 34 min and reduced to 19 min at 45 ˚C (*Figure 2—figure supplement 1*). Interestingly, only 50% of bound nucleotides were released at 4 ˚C suggesting an asymmetric opening of both ATP sites (*Figure 2—figure supplement 1*). We next sought to trace the OF-to-IF return by a time-dependent recovery of peptide binding. In the absence of Mg-ATP, IF TmrA$^{EQ}$B displayed fast peptide binding, which was linearly increasing over time, typical of SPA-based peptide-binding assays (*Figure 2e*). As detailed above, substrate binding to OF TmrA$^{EQ}$B was abolished after the IF-to-OF switch. However, we observed a mono-exponential recovery of peptide binding with a half-life of 24 min, reaching the level determined before nucleotide trapping after 60 min (*Figure 2e*). This demonstrates the full reversibility of the conformational cycle, which we can dissect into two single-turnover events. We also examined the time-dependent release of γ-phosphate upon the OF-to-IF transition. Strikingly, only one of the two occluded ATP molecules is hydrolyzed within the single OF-to-IF transition, approaching a final ATP-to-ADP ratio of 1:1 (*Figure 2f*). This illustrates the non-equivalence of the two ATP sites in ATP turnover as shown by the two asymmetric unlocked-return states captured by cryo-EM (*Hofmann et al., 2019*). The rate of phosphate release complies with the overall ATP hydrolysis rate as well as the rates of nucleotide release and peptide re-binding. Taken together, a single ATP turnover and phosphate release at the consensus site launch the OF-to-IF return, restoring the peptide-binding site and allowing nucleotide exchange.

## The IF-to-OF transition triggered by ATP binding is coupled to substrate translocation

To test whether ATP binding leads to a productive translocation event in the IF-to-OF transition, we reconstituted TmrA$^{EQ}$B in liposomes at a lipid-to-protein ratio of 20:1 (w/w) and performed a single-turnover assay with 50 μM C4F peptide (*Figure 3a*), which is well above its $K_D$ value of 6.8 μM

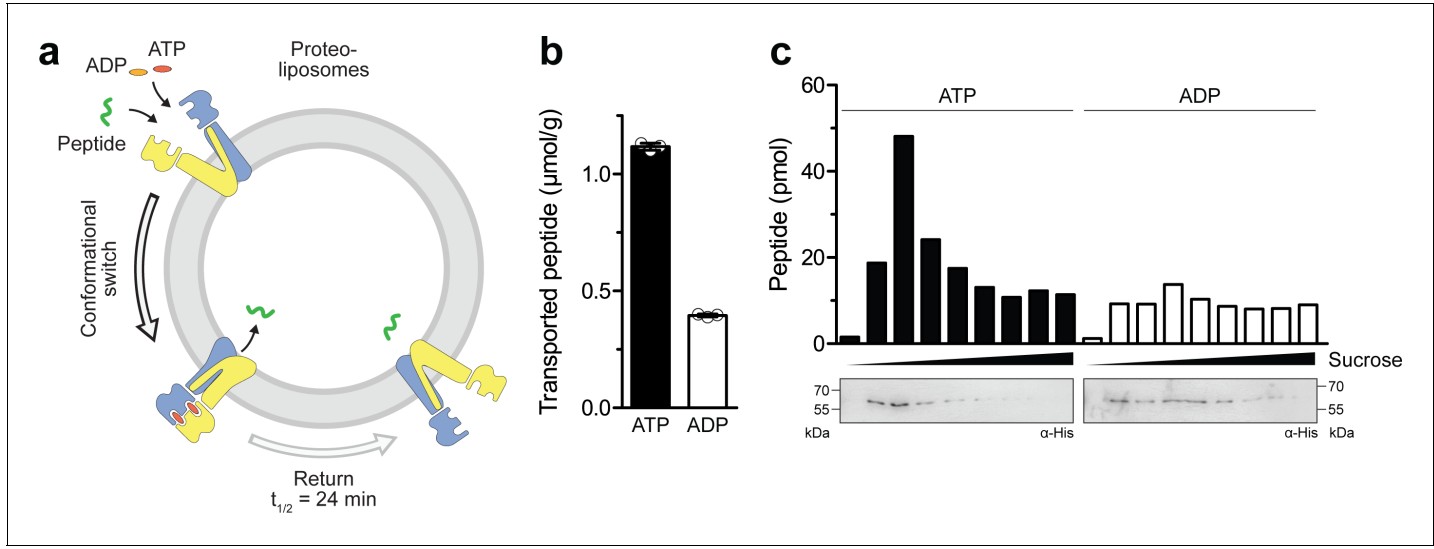

**Figure 3.** Single-conformational switch by ATP binding is coupled to an efficient substrate translocation. (a) Experimental scheme. TmrA$^{EQ}$B was reconstituted in liposomes and incubated with C4F peptide and Mg$^{2+}$-ATP or Mg$^{2+}$-ADP. For single-turnover, transport was performed for 5 min at 45 ˚C to allow a single IF-to-OF transition. After solubilization of proteoliposomes, transported peptides were quantified by fluorescence detection (λ$_{ex/em}$ = 485/520 nm). (b) Single-turnover transport studies. Liposomes containing reconstituted TmrA$^{EQ}$B were mixed with C4F peptide (50 μM) and Mg$^{2+}$-ATP or Mg$^{2+}$-ADP (3 mM each) for 5 min at 45 ˚C. Liposomes were extensively washed. After solubilization, peptides were fluorescently quantified. Data are shown as mean ± SD (n = 3). (c) Single-turnover transport and liposome flotation assays. C4F peptide translocation was performed as described in b). Proteoliposomes were floated by sucrose density gradient centrifugation to remove aggregates and unspecifically adsorbed C4F peptides. Transported peptides were fluorescently quantified. TmrA$^{EQ}$B was detected using immunoblotting (α-His).

The online version of this article includes the following source data and figure supplement(s) for figure 3:

**Source data 1.** Source Data to *Figure 3b and c*.
**Figure supplement 1.** Orientation and incorporation of TmrA$^{EQ}$B in proteoliposomes.

(*Nöll et al., 2017*). When an IF-to-OF transition was induced by ATP binding (5 min, 45 °C), a substantially higher number of proteoliposome-associated peptides was observed as compared to the background level with ADP. Given that TmrA$^{EQ}$B with occluded Mg$^{2+}$-ATP remains arrested in the OF conformation with a half-life of 24 min, these data indicate that ATP binding with the ensuring IF-to-OF switch – and not ATP hydrolysis – represent the power stroke driving substrate translocation (*Figure 3b*). To exclude the possibility that peptides are associated with membranes or protein aggregates in an ATP-dependent manner, we separated liposomes by flotation in sucrose density gradients and quantified the lumenal C4F peptides by fluorescence readout (*Figure 3c*). For proteoliposomes incubated with Mg$^{2+}$-ATP, we observed a substantially higher peptide level compared to Mg$^{2+}$-ADP. Notably, the level of translocated C4F peptides correlated with the amount of TmrA$^{EQ}$B in the floated liposome fraction. In contrast, fractions lacking TmrAB showed only background fluorescence similar to proteoliposomes incubated with ADP, for which the peptide level was independent of the amount of reconstituted TmrA$^{EQ}$B in the corresponding fraction. To determine the coupling ratio of transported peptides per TmrA$^{EQ}$B complex, we characterized proteoliposomes based on (i) TmrA$^{EQ}$B orientation, (ii) incorporation efficiency, and (iii) liposome diameter (*Figure 3—figure supplement 1*). Firstly, the orientation of TmrA$^{EQ}$B in liposomes was determined by TEV-protease cleavage of the His-tag fused to TmrA. As monitored via anti-His immunoblotting, 50% of the His-tags were cleaved, reflecting TmrAB complex with the right-side-out orientation while equal loading of proteoliposomes was controlled by instant blue staining (*Figure 3—figure supplement 1*). Secondly, the reconstitution efficiency was determined by carbonate crush and flotation in sucrose density gradients. Hereby, 95% of TmrA$^{EQ}$B complexes were effectively reconstituted in proteoliposomes and were detected in the top fractions as quantified by anti-His immunoblotting (*Figure 3—figure supplement 1*). Thirdly, the average proteoliposomes diameter (~160 nm) was measured by single-particle tracking (*Figure 3—figure supplement 1*). Finally, we calculated for each liposome ~30 correctly-oriented, energizable transport complexes. Notably, based on the number of translocated peptides per liposome, a nearly stoichiometric coupling of ~0.5 translocated peptides per TmrA$^{EQ}$B complex was calculated.

## ATP binding drives substrate translocation monitored by single-liposome assays

As demonstrated above, ATP binding induces the IF-to-OF transition, which is tightly coupled to peptide translocation. To reveal insights into this single-transition event, we established a single-liposome transport assay with high signal-to-background sensitivity. For each liposome, the number of translocated peptides was correlated with the number of right-side-out reconstituted TmrAB, which was stoichiometrically labeled with *tris*NTA$^{AlexaFluor647}$ (*tris*NTA$^{AF647}$) via the C-terminal His-tag of TmrA as previously described (*Wieneke and Tampé, 2019*). To set up this new approach, we first analyzed substrate translocation by wildtype TmrAB (*Figure 4a*). After incubation with ATP for 5 min at 45 °C, a 100-fold increase in transported peptides was observed as compared to the background control with ADP (*Figure 4b*). It is important mentioning that substrate transport was observed for 98% of TmrAB-positive liposomes. To convert the mean fluorescence intensities into the absolute number of translocated peptides, we calibrated the flow cytometry analysis with liposomes loaded with defined numbers of C4F peptide and equal concentrations of dextran$^{AF647}$ as loading control. We observed a linear increase of the mean fluorescence intensity over the range of 1 to 100 peptides per liposome (*Figure 4—figure supplement 1*).

We next followed peptide translocation by TmrA$^{EQ}$B under single-turnover conditions and quantified the chemomechanical coupling (*Figure 5a*). Proteoliposomes incubated with ATP showed a substantial increase in fluorescent peptide compared to the background level with ADP (*Figure 5b*). Peptide translocation by TmrA$^{EQ}$B showed a biphasic time dependency characterized by a fast burst phase (pre-steady state) and a much slower linear steady-state rate (*Figure 5c*). Within the first minutes, the transported peptides increased until all TmrA$^{EQ}$B complexes had been switched and arrested in the pre-hydrolytic state (burst phase). With longer incubation periods (steady-state condition), a very low translocation rate reflecting multiple ATP turnover events was observed. In the burst phase (0–5 min), however,~30 peptides are transported per liposome (y-axis intercept of the linear term), reflecting the number of transport-competent TmrA$^{EQ}$B complexes per liposome (*Figure 3—figure supplement 1*).

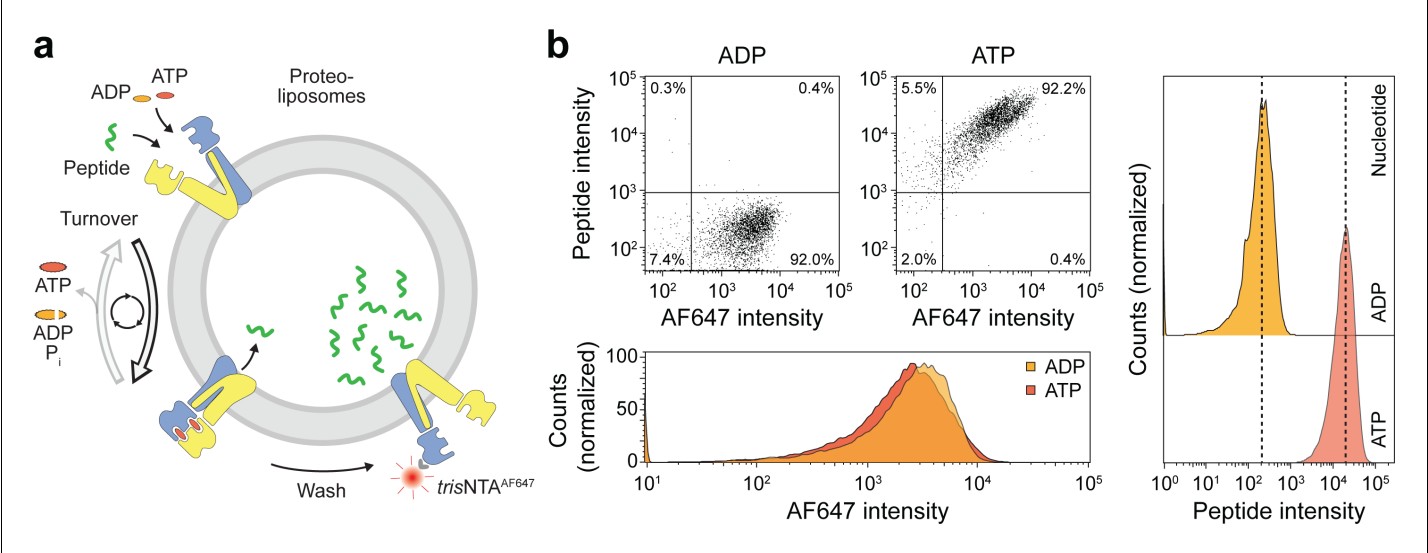

**Figure 4.** Sequential rounds of peptide translocation monitored by flow cytometry. (**a**) Experimental scheme. TmrAB-containing liposomes were incubated with C4F peptide (30 µM) and $Mg^{2+}$-ATP or $Mg^{2+}$-ADP (3 mM each) for 5 min at 45 °C. Proteoliposomes were extensively washed by centrifugation, and His-tagged TmrAB complexes were labeled with trisNTA$^{AF647}$ (150 nM). The mean fluorescence intensities of correctly oriented TmrAB complexes and translocated peptides were examined by flow cytometry. (**b**), Multiple rounds of C4F peptide translocation. Proteoliposomes were incubated with C4F peptide and nucleotides as described in a). Proteoliposomes were washed and correctly-oriented TmrAB complexes were labeled with trisNTA$^{AF647}$ as described in a). The mean fluorescence intensities of 100,000 events were analyzed.

The online version of this article includes the following source data and figure supplement(s) for figure 4:

**Figure supplement 1.** Correlation of the lumenal peptide amount and fluorescence intensity by quantitative flow cytometry.

**Figure supplement 1—source data 1.** Source Data to *Figure 4—figure supplement 1*.

## Single translocation events are unidirectional and stoichiometrically coupled

We next investigated sequential cycles of substrate translocation induced by ATP binding (*Figure 5a*). After the first IF-to-OF switch (switch 1, 5 min at 45 °C), we gave the transport complexes sufficient time to return to the IF conformation (60 min at 4 °C, see *Figure 2*). We then performed a second IF-to-OF transition under single-turnover conditions (switch 2). Proteoliposomes displayed an additive accumulation of peptides along each IF-to-OF switch, while the number of translocated peptides per liposome did not change during each OF-to-IF return, indicating that peptides are not retro-translocated during the OF-to-IF transition (*Figure 5d*). We calculated that ~1.3 peptides are translocated per correctly oriented TmrA$^{EQ}$B upon each IF-to-OF transition. To follow the fate of the substrate after its first translocation, we performed a first single-turnover with an Alexa Fluor 647-labeled peptide (C4$^{AF647}$), followed by a second single-turnover experiment with the C4F peptide (*Figure 5—figure supplement 1*). In the presence of ATP, proteoliposomes displayed elevated signals for both peptides compared to proteoliposomes incubated with ADP. Notably, during the second IF-to-OF switch, the fluorescence signal of C4$^{AF647}$ remained constant, demonstrating that substrate transport occurs unidirectionally. Taken together, sequential rounds of ATP binding drive a unidirectional and stoichiometrically coupled substrate translocation via a single IF-to-OF switch. In turn, ATP hydrolysis and phosphate release in the canonical site is used to master the directionality of the ABC transporter.

## Discussion

In this work, we advance the quantitative mechanistic understanding of ABC exporters. We reveal that ATP binding, which initiates the IF-to-OF transition, is stoichiometrically coupled to substrate translocation (*Figure 6*). Along this power stroke by occlusion of two ATP molecules, the substrate-binding site is remodeled, priming substrate release on the extracellular site. In contrast, ATP

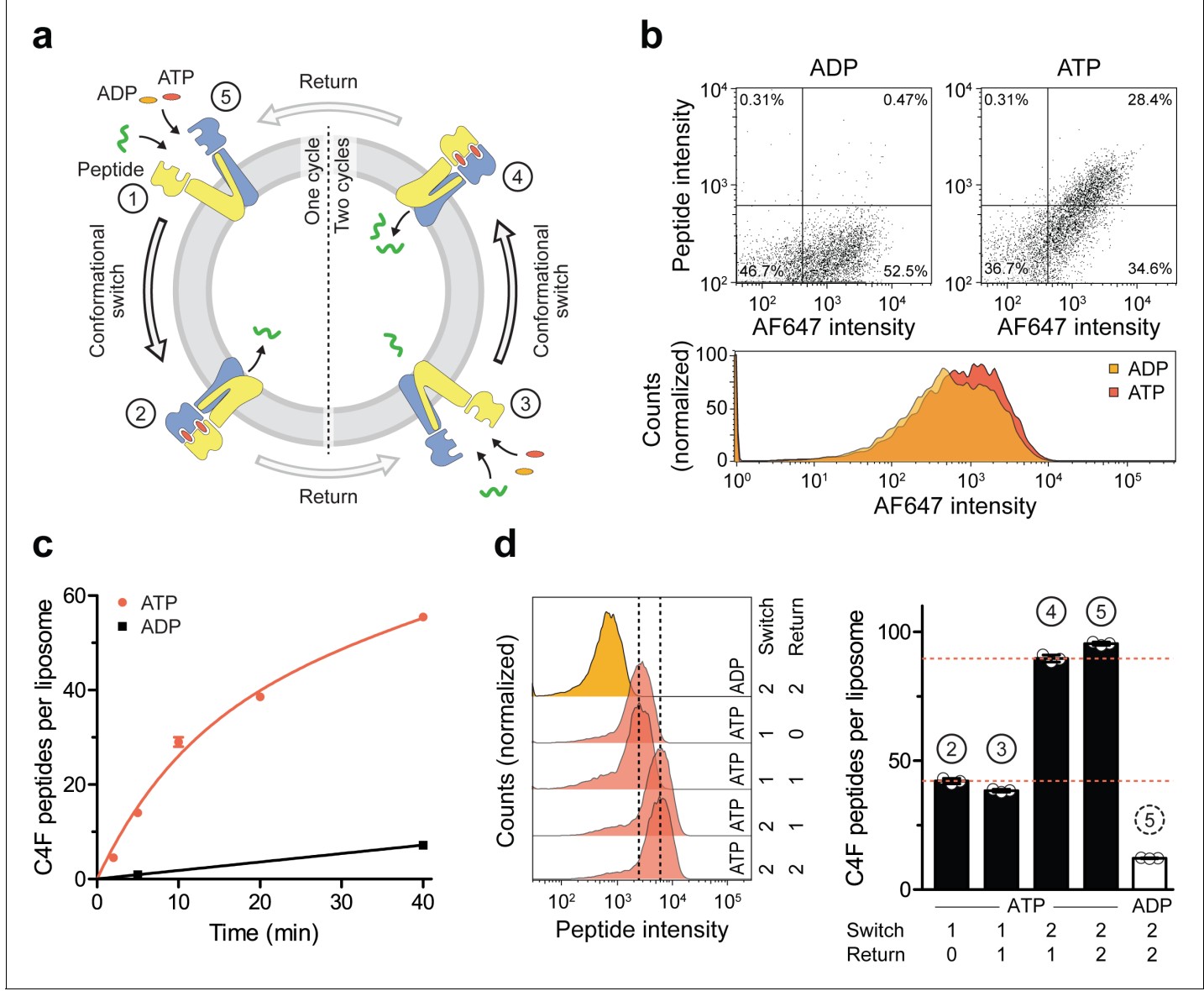

**Figure 5.** A single IF-to-OF transition driving substrate translocation. (a) Experimental setups. Liposomes containing reconstituted TmrA$^{EQ}$B were incubated with C4F peptide (30 μM) and Mg$^{2+}$-ATP or Mg$^{2+}$-ADP (3 mM each). To guarantee a single-turnover, transport was performed for 5 min at 45 ˚C (switch 1). For two consecutive cycles, proteoliposomes were incubated for 1 hr at 4 ˚C (return) followed by 5 min at 45 ˚C (switch 2). Proteoliposomes were extensively washed, and His-tagged TmrA$^{EQ}$B complexes were labeled with *tris*NTA$^{AF647}$ (150 nM). Proteoliposomes were washed by centrifugation. The mean fluorescence intensities of correctly oriented TmrA$^{EQ}$B complexes and translocated peptides were examined by flow cytometry. (b) Single-translocation events monitored by flow cytometry. TmrA$^{EQ}$B proteoliposomes were incubated with C4F peptide and nucleotide as described in a). The mean fluorescence intensities of 100,000 events were analyzed by flow cytometry. (c) Time-dependent translocation of peptides induced by ATP binding and IF-to-OF switching. TmrA$^{EQ}$B proteoliposomes were incubated with C4F peptide and nucleotides as described in a) for increasing time at 45 ˚C. For ATP, signals were subtracted by mean fluorescence intensities of untreated proteoliposomes and fitted by a burst equation (*Equation 6*), yielding ~35 peptides per liposome at the Y-intercept (Span$_{fast}$) and $k_{slow}$ ≈ 0.5 peptides liposome$^{-1}$ min$^{-1}$. Data are shown as mean ± SD (n = 3). (d), Two consecutive rounds of C4F peptide translocation driven by ATP binding and IF-to-OF switching. TmrA$^{EQ}$B proteoliposomes were incubated with C4F peptide and nucleotides as described in a). The mean fluorescence intensities of 10$^5$ events were analyzed by flow cytometry. Data are shown as mean ± SD (n = 3).

The online version of this article includes the following source data and figure supplement(s) for figure 5:

**Source data 1.** Source Data to *Figure 5c and d* as well as *Figure 5—figure supplement 1*.

**Figure supplement 1.** Single-turnover translocation of diverse substrates induced by ATP binding and IF-to-OF switching.

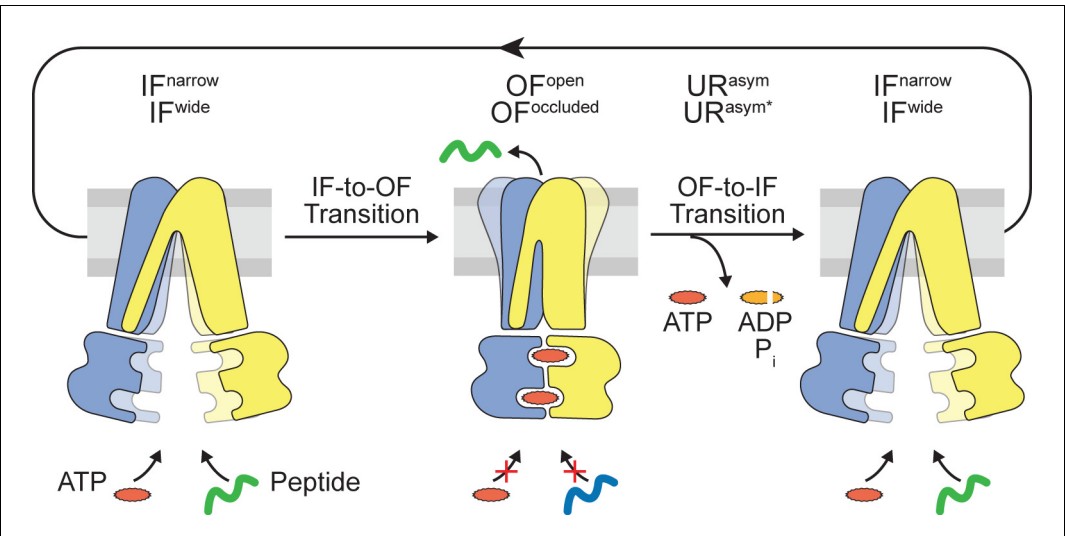

**Figure 6.** Translocation cycle. IF$^{wide}$ conformation of TmrAB allows binding of nucleotides and peptide. ATP binding triggers the IF-to-OF conformational transition, which drives substrate translocation. ATP-occluded OF conformations represent pre-hydrolysis states, in which peptide binding and nucleotide exchange is abolished. After ATP hydrolysis at the consensus site, phosphate (P$_i$) release triggers the OF-to-IF transition via asymmetric unlocked-return (UR$^{asym}$ and UR$^{asym*}$) conformers, revealed by cryo-EM (*Hofmann et al., 2019*), to reactivate peptide and nucleotide binding.

hydrolysis and subsequent phosphate release at the consensus site trigger the OF-to-IF return, which restores substrate binding and nucleotide exchange to start a new translocation cycle. By using equilibrium ligand-binding assays, we demonstrate that the IF-to-OF transition initiated by ATP binding abolishes peptide binding.

The use of a catalytically less active variant in combination with Mg-ATP has been described as an effective strategy for colonizing ABC exporters in the pre-hydrolytic OF state (*Hofmann et al., 2019*; *Kim and Chen, 2018*). Based on this data, we employed the TmrA$^{EQ}$B variant with a slow-down in ATP turnover to kinetically outline the major conformational transitions (IF-to-OF and return) and to correlate these rearrangements with substrate translocation. This approach enabled us to resolve substrate transport in single-turnover mode. Regarding wildtype TmrAB, single-turnover transport studies would require conformation-selective inhibitors which arrest the transport complex at different stages of the translocation cycle and which to our knowledge are not available.

A comparison of IF and OF TmrAB points to a drastic reorganization of residues aligning the substrate-binding site (*Hofmann et al., 2019*), similar to P-glycoprotein (*Martin et al., 2000*; *Martin et al., 2001*), ABCG2 (*McDevitt et al., 2008*), and MRP1 (*Payen et al., 2003*). In addition, we delineate that the OF-to-IF return directly correlates with (i) the ATP turnover, (ii) the release of inorganic phosphate, (iii) the restoration of peptide binding, and (iv) nucleotide exchange. We observed a full recovery of peptide binding, demonstrating reversibility of conformational transitions to start subsequent processing cycles.

The processive-clamp/ATP-switch model (*Abele and Tampé, 2004*; *Higgins and Linton, 2004*; *Janas et al., 2003*) has gained increasing acceptance in the light of new structures of ABC transporters solved in the nucleotide-bound state (*Johnson and Chen, 2018*; *Manolaridis et al., 2018*). However, because of major experimental demands, it is an open debate whether the substrate translocation event is driven by ATP binding or ATP hydrolysis. We addressed this pressing question by establishing a sensitive single-turnover translocation assay. We uncovered that ATP binding drives the single-translocation event of different substrates along the IF-to-OF transition. By analyzing sequential steps in the translocation cycle, we revealed that substrate transport by TmrA$^{EQ}$B is unidirectional, which was previously shown for human TAP (*Grossmann et al., 2014*).

In ABC transporters, the coupling ratio between conformational transitions and substrate translocation remain largely elusive. The coupling between ATP hydrolysis and the transported substrate

can be quite ineffective, reaching up to 100 ATP-turnover events per translocated substrate (**Bock et al., 2019**; **Manolaridis et al., 2018**; **Nöll et al., 2017**; **Zehnpfennig et al., 2009**). Experimental parameters such as detergents, lipid environment, sample preparation, protein activity, and sensitivity of applied methods can substantially affect the energetic coupling. Here, our newly established single-turnover approaches reveal a coupling ratio of 0.5 to 1.3 peptides per TmrA$^{EQ}$B. Taking the experimental error and sensitivity of these approaches into account, we revealed a stoichiometric coupling between a single conformational transition and a translocation event. Sensitive translocation assays with single-liposome or single-transporter resolution will be indispensable to elucidate the main factors that control and affect the chemomechanical coupling.

Our new single-liposome approaches revealed a fast burst phase followed by a slower steady-state phase, indicating that a step after substrate translocation is rate-limiting. In our cryo-EM structures of TmrAB, we captured two asymmetric ADP/ATP-bound states under steady-state conditions, leading to the assumption that the separation of the NBD dimer represents the rate-limiting step (**Hofmann et al., 2019**). Here, we demonstrate that the rates of phosphate, ATP, and ADP release correlate with the OF-to-IF transition. In the future, single-molecule fluorescence resonance energy transfer (**Catipovic et al., 2019**), time-resolved small-angle X-ray scattering (**Josts et al., 2020**), and stopped-flow analyses (**Robson et al., 2009**) could provide further insights into turnover kinetics and rate-limiting steps.

In conclusion, we delineate the mechanistic framework of major conformational transitions and the fundamental energetic principles in heterodimeric ABC transporters. Our single-turnover analyses significantly advance our quantitative mechanistic understanding how transitional rearrangements in ABC transport complexes master substrate binding and translocation. We uncovered that the IF-to-OF transition by ATP binding drives unidirectional substrate translocation across the membrane. In the future, single-turnover approaches will be an indispensable method to unfold hidden facets of conformational coupling, energy transduction, and mechanistic details of substrate translocation by ABC transporters but also other membrane translocation machineries.

# Materials and methods

## Key resources table

| Reagent type (species) or resource | Designation | Source or reference | Identifiers | Additional information |
|---|---|---|---|---|
| Antibody | Monoclonal α-His | Sigma-Aldrich | SAB1305538 | Mouse origin; 1:2000 dilution |
| Antibody | Polyclonal α-mouse-HRP conjugate | Sigma-Aldrich | AP130P | Goat origin; 1:20,000 dilution |
| Chemical compound, drug | β-n-dodecyl β-D-maltoside (DDM) | Carl Roth | CN26.5 | |
| Chemical compound, drug | Bovine brain lipid extract | Sigma-Aldrich | B1502 | |
| Chemical compound, drug | *E. coli* polar lipids | Avanti | 100600 | |
| Chemical compound, drug | 1,2-Dioleoyl-sn-Glycero-3-Phosphocholine (DOPC) | Anatrace | D518 | |
| Chemical compound, drug | [2,5′,8-$^3$H(N)]-ATP ($^3$H-ATP) | PerkinElmer | NET118900 | |
| Chemical compound, drug | [2,8-$^3$H]-ADP ($^3$H-ADP) | Biotrend | ART-0386 | |
| Chemical compound, drug | [α$^{32}$P]-ATP | Hartmann Analytic | FP-207 | |
| Chemical compound, drug | [γ$^{32}$P]-ATP | Hartmann Analytic | FP-201 | |
| Chemical compound, drug | Copper-chelated PVT SPA beads | PerkinElmer | RPNQ0095 | |

*Continued on next page*

*Continued*

| Reagent type (species) or resource | Designation | Source or reference | Identifiers | Additional information |
|---|---|---|---|---|
| Gene (*Thermus thermophilus*) | TmrA | Q72J05 | TTC0976 | |
| Gene (*Thermus thermophilus*) | TmrB | Q72J04 | TTC0977 | |
| Peptide, recombinant protein | RRY-[$^3$H]L-KSTEL | Hartmann Analytic | Custom synthesis | |
| Peptide, recombinant protein | RRY-C*-KSTEL | This study (Peptide labeling and purification) | | C* denotes fluorescein labeled Cys |
| Peptide, recombinant protein | RRY-C*-KSTEL | This study (Peptide labeling and purification) | | C* denotes Alexa Fluor 647 labeled Cys |
| Recombinant DNA reagent | pET-22b | Merck Millipore | 69744 | Vector for protein expression in *E. coli* |
| Software, Algorithm | Prism 5 | GraphPad | | |
| Software, Algorithm | PyMOL | Schrödinger | | |
| Strain, strain background (*Escherichia coli*) | BL21(DE3) | Thermo Fisher | C600003 | Chemically competent cells |

## Expression and purification of TmrAB

Expression and purification of TmrAB was performed as described (*Hofmann et al., 2019*; *Zutz et al., 2011*). TmrAB was expressed in *E. coli* BL21(DE3) grown in LB high salt media at 37 ˚C. At an $OD_{600}$ of 0.6, protein expression was induced with 0.5 mM isopropyl β-D-thiogalactopyranoside (IPTG, Carbolution). Cells were grown for 3 hr at 37 ˚C. Harvested cells were resuspended in lysis buffer (20 mM HEPES-NaOH pH 7.5, 300 mM NaCl, 50 µg/ml lysozyme, 0.2 mM PMSF) and disrupted by sonication. Membranes were pelleted at 100,000 g for 30 min at 4 ˚C. Membranes were solubilized for 1 hr at 4 ˚C with 20 mM β-n-dodecyl β-D-maltoside (β-DDM, Carl Roth) in purification buffer (20 mM HEPES-NaOH pH 7.5, 300 mM NaCl). After centrifugation at 100,000 g for 30 min at 4 ˚C, solubilized TmrAB was applied to Ni-NTA agarose (Qiagen) for 1 hr at 4 ˚C. Resin was washed with 20 column volumes of purification buffer containing 50 mM imidazole and 1 mM β-DDM. Bound protein was eluted with purification buffer including 300 mM imidazole and 1 mM β-DDM. The eluate was buffer exchanged to TSK buffer (20 mM HEPES-NaOH pH 7.5, 150 mM NaCl, 1 mM β-DDM) using PD-10 desalting column (GE Healthcare).

## Expression and purification of membrane scaffolds

MSP1D1 was expressed in *E. coli* BL21(DE3) grown in LB high salt media at 37 ˚C (*Hofmann et al., 2019*). Protein expression was induced by addition of 0.5 mM IPTG at an $OD_{600}$ of 0.6. Cells were grown for 3 hr at 37 ˚C. Purification was performed as described (*Ritchie et al., 2009*). Briefly, harvested cells were resuspended in 40 mM Tris-HCl pH 8.0, 300 mM NaCl, 1 mM PMSF and 1% Triton X-100. Cells were disrupted using sonication, and lysate was cleared at 30,000 g for 30 min at 4 ˚C. MSP1D1 was applied to Ni-NTA agarose (Qiagen) for 1 hr at 4 ˚C. Resin was washed consecutively with 20 column volumes of Tris-HCl buffer (40 mM Tris-HCl pH 8.0, 300 mM NaCl), containing (i) 1% Triton X-100, (ii) 50 mM Na-cholate, 20 mM imidazole, and (iii) 50 mM imidazole. Bound MSP1D1 was eluted in Tris-HCl buffer containing 400 mM imidazole. Eluate was buffer exchanged to 20 mM Tris-HCl pH 7.4, 100 mM NaCl and 0.5 mM EDTA using PD-10 desalting column (GE Healthcare).

## Reconstitution in lipid nanodiscs

Reconstitution of TmrAB in lipid nanodiscs was performed as described (*Hofmann et al., 2019*). Bovine brain lipids (Sigma-Aldrich) were solubilized in 20 mM of β-DDM. Purified TmrAB and

MSP1D1 were mixed with DDM-solubilized bovine brain lipids in a TmrAB/MSP1D1/lipid molar ratio of 1/7.5/100 in TSK buffer without detergent. Samples were incubated at 20 °C for 30 min. SM-2 Bio-beads (Bio-Rad) were added in two steps at 4 °C (1 hr and overnight) to remove detergent. Samples were concentrated using Amicon Ultra-0.5 ml centrifugal filters with 50 kDa cut-off (Merck Millipore). TmrAB reconstituted in lipid nanodiscs were separated from empty nanodiscs by size exclusion chromatography (SEC) using Superdex 200 Increase 3.2/300 (GE Healthcare).

## Liposome reconstitution

Reconstitution of TmrAB in liposomes was performed as described (Nöll et al., 2017). Briefly, liposomes were composed of *E. coli* polar lipids/DOPC (Anatrace) in a 7/3 (w/w) ratio. Purified TmrAB was added to Triton X-100 destabilized liposomes in a 1/20 (w/w) ratio. SM-2 Bio-beads (Bio-Rad) were added in four steps at 4 °C (1 hr, overnight, 2 hr, and 2 hr) to remove the detergent. Proteoliposomes were harvested at 270,000 g for 30 min at 4 °C and resuspended in transport buffer (20 mM HEPES-NaOH pH 7.5, 150 mM NaCl, 5% (v/v) glycerol) to a final lipid concentration of 5 mg/ml. TmrAB orientation in liposomes was examined using TEV protease specific cleavage of the C-terminal His-tag. Briefly, liposomes were mixed with TEV protease in a TmrAB/TEV protease ratio of 16/1 (w/w). Digestion was performed for 16 hr at 4 °C including 3 mM DTT. As positive control, proteoliposomes were solubilized during digestion with 1% (w/v) Triton X-100. His-tag cleavage of TmrAB was quantified by immunoblotting using α-His detection (Sigma-Aldrich). An equal load of TmrAB was confirmed by SDS-PAGE and InstantBlue staining (Sigma-Aldrich). The reconstitution of TmrAB in liposomes was analyzed by carbonate crush and sucrose density gradient flotation assays. Proteoliposomes were harvested at 270,000 g for 30 min at 4 °C and resuspended in 100 mM $Na_2CO_3$ pH 11.5, 30% (w/v) sucrose. Samples were incubated for 30 min on ice. Liposomes were layered with HEPES-NaOH buffer (20 mM HEPES-NaOH pH 7.5, 150 mM NaCl) containing (i) 30% sucrose, (ii) 20% sucrose and (iii) 0% sucrose. Samples were centrifuged at 270,000 g for 30 min at 4 °C. The gradient was fractionated, and TmrAB was quantified by immunoblotting using α-His antibody (Sigma-Aldrich). The size of the proteoliposomes were determined by the nanoparticle tracking system NanoSight LM10 (Malvern Panalytical). Reconstituted proteoliposomes were diluted to a lipid concentration of 1 mg/l with transport buffer, and vesicle tracking was performed at 25 °C.

## Peptide binding assays

Equilibrium peptide binding to TmrAB was examined using scintillation proximity assays (SPA). TmrAB (0.2 µM) reconstituted in lipid nanodiscs was mixed with the peptide RRYQKSTEL (9.75 µM of R9LQK) traced with tritylated RRY-[$^3$H]L-KSTEL (0.25 µM of $^3$H-R9L, Hartmann Analytic). To induce the IF-to-OF transition, TmrA$^{EQ}$B was incubated with 1 mM ATP (Sigma-Aldrich) and 3 mM $MgCl_2$ for 5 min at 45 °C. If indicated, the excess of free ATP and peptide was removed by rapid gel filtration (Bio-Spin columns P-30, Bio-Rad). Copper-chelated SPA beads (PerkinElmer) were added to a final concentration of 5 mg/ml. The scintillation proximity assay was performed at 20 °C in cpm mode (Wallac MicroBeta). Background was determined in the presence of 200 mM imidazole. Data represent mean ± SD from three experiments. The total binding signal was background-corrected. The specific binding was multiplied by the dilution factor of radiolabeled $^3$H-R9L. The recovery of peptide binding during the OF-to-IF transition was followed by mono-exponential fit

$$Y = Y_0 + (Plateau - Y_0) \cdot \left(1 - e^{kt}\right) \tag{1}$$

with Y: binding, $Y_0$: binding at X = 0, *k*: rate constant. The half-life was calculated as

$$t_{1/2} = \frac{ln2}{k} \tag{2}$$

Peptide binding was analyzed by a one-site Langmuir-type binding model

$$Y = \frac{B_{max} \cdot [P]}{K_D + [P]} \tag{3}$$

with Y: binding, $B_{max}$: maximum binding, and $K_D$: equilibrium dissociation constant.

## Nucleotide binding assays

Nucleotide binding to TmrAB was examined using SPA. 200 nM TmrAB reconstituted in lipid nanodiscs were incubated with 2.8 µM ATP supplemented with 0.2 µM of [2,5′,8-$^3$H(N)]-ATP ($^3$H-ATP, PerkinElmer) or 2.8 µM ADP supplemented with 0.2 µM of [2,8-$^3$H]-ADP ($^3$H-ADP, Biotrend), and 3 mM MgCl$_2$. If indicated, the excess of free nucleotides was removed by rapid gel filtration (Bio-Spin columns P-30, Bio-Rad). In some cases, rebinding of occluded nucleotides was blocked by adding an excess of ATP (1 mM). Samples were kept on ice to prevent ATP turnover and ATP-induced conformational changes. Nucleotide binding was analyzed by a one-site binding model (analog to *Equation 3*). Due to excessive background signals, the nucleotide concentration could not be increased above 0.3 mM. Data represent mean ± SD from three experiments.

## Fluorescence anisotropy

Binding of fluorescent peptide RRYC$^{fluorescein}$KSTEL (C4F) was analyzed as described (*Nöll et al., 2017*). Briefly, TmrAB (1 µM) reconstituted in lipid nanodiscs was incubated for 5 min at 45 ˚C in the presence or absence of 1 mM ATP and 3 mM MgCl$_2$ in 20 mM HEPES-NaOH pH 7.5, 150 mM NaCl. In some cases, ATP and ADP were titrated. The fluorescein anisotropy of the C4F peptide (50 nM) was analyzed at $\lambda_{ex/em}$ of 485/520 nm using a microplate reader (CLARIOstar, BMG LABTECH). The fluorescence anisotropy was calculated using

$$r = \frac{I_{||} - I_{\perp}}{I_{||} + 2 \cdot I_{\perp}}$$

(4)

Data represent mean ± SD from three experiments. To determine nucleotide effects on peptide binding, fluorescence anisotropy was normalized to free and 100%-bound C4F peptide and fitted by dose-response function

$$Y = Bottom + \frac{Top - Bottom}{1 + 10^{(X - logEC_{50})}}$$

(5)

with Y: normalized binding, Bottom and Top: plateau values of regression function, EC$_{50}$: effective concentration 50%, X: log[nucleotide].

## ATPase activity assays

TmrA$^{EQ}$B (1 µM) reconstituted in lipid nanodiscs was incubated with 1 mM ouabain, 5 mM NaN$_3$, 50 µM EGTA, 3 mM MgCl$_2$, and 1 mM ATP traced with [γ$^{32}$P]-ATP (Hartmann Analytic). Background hydrolysis was recorded in the presence of 10 mM EDTA and in the absence of TmrAB. Samples were incubated for 5 min at 45 ˚C and 15 min at 20 ˚C to mimic the trapping procedure of TmrA$^{EQ}$B. Samples were spotted onto polyethyleneimine cellulose plates (Merck Millipore). Thin layer chromatography was performed with 0.8 M LiCl-acetic acid pH 3.2. Plates were developed overnight on Exposure Cassette-K (Bio-Rad) and evaluated on Personal Molecular Imager System (Bio-Rad). Due to time delays between nucleotide trapping, thin layer chromatography and autoradiography, background levels of ADP resulting from autohydrolysis could not be avoided. Data represent mean ± SD from three experiments.

## Nucleotide occlusion

After nucleotide trapping of TmrA$^{EQ}$B, occluded nucleotides were analyzed as described (*Zutz et al., 2011*). In brief, TmrA$^{EQ}$B (2 µM) reconstituted in lipid nanodiscs was incubated with 3 mM MgCl$_2$ and 1 mM ATP traced with [γ$^{32}$P]-ATP or [α$^{32}$P]-ATP (Hartmann Analytic) for 5 min at 45˚C. After arresting TmrA$^{EQ}$B in the OF conformation, 2 mM of cold ATP were added and free radiolabeled nucleotides were removed by rapid gel filtration (Bio-Spin columns P-30, Bio-Rad). The nucleotide-trapped TmrA$^{EQ}$B was directly transferred into 2 mM ATP at 20 ˚C to follow a single-turnover. At different time points, samples were spotted onto polyethyleneimine cellulose plates (Merck Millipore). Thin layer chromatography was performed with 0.75 M KH$_2$PO$_4$ pH 3.4. Plates were developed overnight on Exposure Cassette-K (Bio-Rad) and evaluated on Personal Molecular Imager System (Bio-Rad). Data represent mean ± SD from three experiments.

## Peptide transport assays

TmrA$^{EQ}$B (1.5 µM) reconstituted in liposomes was incubated with 50 µM C4F peptide, 3 mM ATP or ADP, and 5 mM MgCl$_2$. Transport was performed for 5 min at 45 ˚C and stopped with 10 mM EDTA. Proteoliposomes were washed and centrifuged three times for 30 min at 270,000 g, 4 ˚C in 20 mM HEPES-NaOH pH 7.5, 150 mM NaCl, 5% (v/v) glycerol, and 10 mM EDTA. Afterwards, proteoliposomes were either (i) solubilized with 1% (w/v) SDS, and peptides were quantified at $\lambda_{ex/em}$ of 485/520 nm using microplate reader, or (ii) applied to a sucrose density gradient flotation assay to remove any peptide bound to liposomes or aggregated TmrA$^{EQ}$B. For this, liposomes were resuspended in 20 mM HEPES-NaOH pH 7.5, 150 mM NaCl, and 55% (w/v) sucrose. The liposome fraction was covered with layers of buffer with decreasing concentrations of sucrose (55% to 0%). Liposomes were separated at 100,000 g for 16 hr at 4 ˚C. The gradient was harvested in fractions. TmrAB was quantified by immunoblotting using an α-His antibody (Sigma-Aldrich). Fractions were solubilized with 1% (w/v) SDS, and peptides were quantified at $\lambda_{ex/em}$ of 485/520 nm using a microplate reader. Data represent mean ± SD from three experiments.

## Single liposome flow cytometry

Wildtype TmrAB or TmrA$^{EQ}$B (0.6 µM) reconstituted in liposomes was incubated with 30 µM C4F peptide, 3 mM ATP or ADP, and 5 mM MgCl$_2$. To probe for single-turnover translocation, TmrA$^{EQ}$B was incubated for 5 min at 45 ˚C (switch 1). For two consecutive cycles, proteoliposomes were kept for 1 hr at 4 ˚C (return) to enable the OF-to-IF transition. Samples were incubated for 5 min at 45 ˚C (switch 2) followed by the second return. For dual-substrate translocation, the first switch was performed with 30 µM C4$^{AF647}$ (Alexa Fluor 647). ADP (20 mM) was added for 1 hr at 4 ˚C to enable the OF-to-IF transition and to prevent additional IF-to-OF rearrangements (return). Proteoliposomes were washed twice by centrifugation (270,000 g, 30 min) and 30 µM C4F peptide, 3 mM ATP or ADP, and 5 mM MgCl$_2$ were added for 5 min at 45 ˚C (switch 2). Proteoliposomes were washed by centrifugation. His-tagged TmrA$^{EQ}$B complexes were labeled with 0.15 µM $tris$NTA$^{AF647}$ for 30 min at 4˚C, and proteoliposomes were washed again. Mean fluorescence intensities (MFIs) of correctly inserted TmrA$^{EQ}$B complexes and translocated substrates were analyzed by flow cytometry (FACS-Celesta). In order to convert the mean fluorescence intensities of lumenal peptides into the number of transported peptides, a regression function was determined. Briefly, liposomes were destabilized with Triton X-100 in the presence of increasing C4F peptide concentrations. Equal portions of dextran$^{AF647}$ served as loading control. Detergent was removed by the addition of SM-2 Bio-beads (Bio-Rad) as detailed above. Liposomes were harvested and washed three times at 270,000 g for 30 min at 4 ˚C. Generally, 10$^5$ proteoliposomes were selected according to sideward and forward scatter areas. Single proteoliposomes were gated based on the height of forward scatter plotted against the area of forward scatter. AF647 positive events were selected, and the substrate intensities were calculated using FlowJo 10.6.1 software. To determine the time dependency of peptide translocation, data were fitted by a burst equation

$$Y = Span_{fast} \cdot \left(1 - e^{-k_{fast} \cdot t}\right) + k_{slow} \cdot t \tag{6}$$

with Span$_{fast}$: intercept of y-axis, $k_{fast}$: rate constant of burst phase, $k_{slow}$: slope of linear steady-state phase. Data were recorded in triplicates and mean ± SD are displayed.

## Peptide labeling and purification

Peptides were synthesized on solid phase by standard Fmoc chemistry. For cysteine labeling, the C4 peptide (RRYCKSTEL) was incubated with 5-iodoacetamide fluorescein (Sigma-Aldrich) or Alexa Fluor 647 maleimide (Thermo Fisher) in PBS DMF buffer (8.1 mM Na$_2$HPO$_4$ pH 6.5, 137 mM NaCl, 2.7 mM KCl, 1.8 mM KH$_2$PO$_4$, 20% (v/v) DMF) for 1 hr at 20 ˚C using a molar fluorophore excess of 1.2. Samples were purified by reversed-phase HPLC (Jasco; PerfectSil 300 ODS C$_{18}$) applying a linear acetonitrile gradient from 5–80% supplemented with 0.1% (v/v) TFA. Purified peptides were snap frozen in liquid nitrogen and lyophilized (Lyovac GT2, Heraeus).

## Data presentation and statistics

All measurements were performed in triplicates ($n$ = 3). In all column and XY diagrams, mean values ± SD were presented. Diagrams were prepared in GraphPad Prism5. Figures of protein structures were prepared using PyMOL (*Schrödinger, 2015*).

## Acknowledgements

We thank Dr. Rupert Abele, Dr. Christoph Thomas, Andrea Pott, Inga Nold, and all members of the Institute for Biochemistry (Goethe University Frankfurt) for discussion and comments. This research was supported by the German Research Foundation (SFB 807 – Membrane Transport and Communication as well as Ta157/12-1 to RT). The support by the European Research Council (ERC Advanced Grant 789121 to RT) is gratefully acknowledged.

## Additional information

### Funding

| Funder | Grant reference number | Author |
| --- | --- | --- |
| Deutsche Forschungsge-meinschaft | Ta157/12-1 | Robert Tampé |
| European Research Council | ERC_Ad 789121 | Robert Tampé |
| Deutsche Forschungsge-meinschaft | SFB 807 | Robert Tampé |

The funders had no role in study design, data collection and interpretation, or the decision to submit the work for publication.

### Author contributions

Erich Stefan, Conceptualization, Data curation, Formal analysis, Validation, Investigation, Visualization, Methodology, Writing - original draft, Writing - review and editing; Susanne Hofmann, Data curation, Investigation; Robert Tampé, Conceptualization, Data curation, Formal analysis, Supervision, Funding acquisition, Validation, Visualization, Writing - review and editing

### Author ORCIDs

Robert Tampé (iD) https://orcid.org/0000-0002-0403-2160

### Decision letter and Author response

Decision letter https://doi.org/10.7554/eLife.55943.sa1
Author response https://doi.org/10.7554/eLife.55943.sa2

## Additional files

### Supplementary files

• Transparent reporting form

### Data availability

All data generated or analysed during this study are included in the manuscript and supporting files. Source data files have been provided for Figures 1, 2, 3, 4, and 5.

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
