## [Decision Letter]

**Acceptance summary:**

ATP-binding cassette transporters form one of the most important classes of transporters and the mechanisms of substrate binding and translocation have been studied for several decades. This report uses single-turnover analyses on a system reconstituted in liposomes to demonstrate that ATP binding and not ATP hydrolysis is the step which drives the transport of the substrate across the membrane. Such an approach might be highly valuable for other transport systems.

**Decision letter after peer review:**

Thank you for submitting your article "A single power stroke by ATP binding drives substrate translocation in a heterodimeric ABC transporter" for consideration by *eLife*. Your article has been reviewed by three peer reviewers, one of whom is a member of our Board of Reviewing Editors, and the evaluation has been overseen by Olga Boudker as the Senior Editor. The following individuals involved in review of your submission have agreed to reveal their identity: Konstantinos Beis (Reviewer #2); Dirk Slotboom (Reviewer #3).

The reviewers have discussed the reviews with one another and the Reviewing Editor has drafted this decision to help you prepare a revised submission.

All three reviewers like the approach and thoroughness of the work and agree that the results provide solid evidence on the previous assumptions that ATP binding and not hydrolysis drives the export of substrates by ABC transporters. They agree that the authors have setup an elegant system to study substrate transport by ABC transporters in detail and that your work is potentially applicable to any ABC transporter assuming that fluorescent labelled or radioligands are available.

No extra experiments are required but the reviewers asked for several clarifications.

Essential revisions:

None, but I would suggest that you address the points below.

Minor points:

1) The reviewers agree that the use of the EQ mutant is essential for this manuscript, however they would like you to discuss shortly to use the EQ mutant might affect their results.

2) In subsection “The IF-to-OF transition triggered by ATP binding is coupled to substrate translocation” the authors suggests that ATP binding and not hydrolysis drives transport, based on their observation that after 5 minutes incubation they see “a substantially higher number of peptides” transported and their observation that ATP binding is much faster than ATP hydrolysis. As long as the amount is not quantified or placed in relation to the total amount of peptides transported, no difference can be made between full binding or only a partial hydrolysis of ATP. The authors set-out to do this quantification. I would suggest that the authors make their suggestion that the powerstroke is driven by binding and not by hydrolysis after this quantification.

3) In Figure 2F the error bars seem to be missing.

4)The authors state that “each liposome contained ~30 correctly-oriented, energizable transport complexes.” Can they clarify what evidence they have apart from the transport of peptides, about the number of molecules? Have they quantified from their fluorescent labelled protein?

5) Figure 2B could benefit by having a darker contrast.

6) The Results section is extremely condensed and hard to read. Since e*Life* does not have a firm word limit to manuscripts, the authors really should do better in explaining the (nice) experiments. In addition, I find the discussion a bit shallow, and disappointing considering the beautiful data.

7) Subsection “A single ATP turnover and phosphate release trigger the OF-to-IF return”: "as described above" Removal of ATP was not described above.

8) Figure 2D and related supplement figure: The decay does not look monoexponential. I suspect that there are 2 different *k_off_* values: One for the dissociation of nucleotide from the canonical site (after hydrolysis) and the other for dissociation from the degenerate site. In that respect, it is surprising that in Figure 2F the nucleotide from the degenerate site does not seem to come off. Why does more than 80% of bound nucleotide dissociate in the experiment shown in 2D (this number indicates that dissociation from both sites takes place), but only 50% in 2F (this number suggests that ATP in the degenerate site does not come off).

9) Figure 2E: In the "No nucleotide" control there is a clear increase in bound peptide over the time course of the experiment. This must be explained. My guess is that ATP co-purifies with the protein.

10) Figure 2—figure supplement 1 panel B: At 4 degrees, only 50% of bound nucleotide is released. Explain. It may be that the dissociation from the degenerate site is much slower at 4 degrees than at higher temp, or that the dissociation from the canonical site is much slower. To tell these two events apart would really give new insight.

11) Subsection “A single ATP turnover and phosphate release trigger the OF-to-IF return”: The difference in half-life at the different temperatures is very small. Given the large conformational change needed for the dissociation to occur, I find the marginal temperature dependence surprising. Explain.

12) Figure 1D and Figure 1—figure supplement 2 panel B: It is unclear whether these experiments are identical or not. If yes: why show it twice? If not: explain the difference better.

13) Figure 1B and Figure 1—figure supplement 1 panel C: why are the absolute values (cpm) so different?

14) Subsection “ATP binding drives substrate translocation monitored by single-liposome assays”: it is not clear what "untreated" refers to.

15) Figure 3—figure supplement 1 panel A. It is unclear what the panels show, and there is no explanation in the text. Why are the instant blue data different from the α-His data? The samples were incubated for the same amount of time, right? Why is there no quantification shown? From the qualitative images, the authors seem to draw quantitative conclusion. Please show proper quantification.

16) Figure 5C. I do not see biphasic behavior, and do not see the offset at 35 per liposome. I guess the wrong figure is shown?

17) In general, the authors should not oversell their nice work by using words like "ultrasensitive", unscientific qualifications like "almost unlimited". Also, it does not help the reader if it is unclear what "this" and "these" refer to. For instance, in paragraph two of the Introduction "this question": there was no question asked. Paragraph three: “these questions”: again, I cannot find questions.

---

## [Author Response]

Essential revisions:None, but I would suggest that you address the points below.Minor points:1) The reviewers agree that the use of the EQ mutant is essential for this manuscript, however they would like you to discuss shortly to use the EQ mutant might affect their results.

Thank you for your comment. Using a catalytically reduced variant in combination with Mg-ATP appeared as an effective strategy to populate the OF conformation as reported for TmrAB but also for other ABC exporters (Hofmann et al., 2019; Manolaridis et al., 2018; Kim et al., 2018; Dastvan et al., 2019; Barth et al., 2018). In order to investigate the events of ATP binding and ATP hydrolysis with their respective effects on substrate translocation, we utilized the TmrA^EQ^B variant with a drastic slowdown in ATP turnover. This allowed us to kinetically outline the IF-to-OF switch and the returning OF-to-IF transition to correlate these conversions with substrate translocation. Considering wildtype TmrAB, single-turnover transport studies are technically challenging and would require conformation-selective inhibitors that arrest TmrAB at various stages of the translocation cycle which to our knowledge are presently not available. We observed an effective coupling ratio of consumed ATP per translocated substrate for TmrA^EQ^B in contrast to TmrAB which acts largely uncoupled (Bock et al., 2019). As the speed of conformational transitions is slowed down for TmrA^EQ^B compared to TmrAB, the allosteric coupling might be favored. For explanation, we extended the main text accordingly.

2) In subsection “The IF-to-OF transition triggered by ATP binding is coupled to substrate translocation” the authors suggests that ATP binding and not hydrolysis drives transport, based on their observation that after 5 minutes incubation they see “a substantially higher number of peptides” transported and their observation that ATP binding is much faster than ATP hydrolysis. As long as the amount is not quantified or placed in relation to the total amount of peptides transported, no difference can be made between full binding or only a partial hydrolysis of ATP. The authors set-out to do this quantification. I would suggest that the authors make their suggestion that the powerstroke is driven by binding and not by hydrolysis after this quantification.

We like to thank the reviewers for this comment. We have changed the main text accordingly.

3) In Figure 2F the error bars seem to be missing.

The error bars were already displayed in the diagram but hard to read. We have changed the figure accordingly.

4)The authors state that “each liposome contained ~30 correctly-oriented, energizable transport complexes.” Can they clarify what evidence they have apart from the transport of peptides, about the number of molecules? Have they quantified from their fluorescent labelled protein?

Thank you for this helpful comment. We would like to refer to Figure3—figure supplement1. Proteoliposomes were characterized by nanoparticle tracking to calculate the average diameter. TmrA^EQ^B was randomly oriented (50/50%) in proteoliposomes determined by TEV-protease cleavage of the His-tag fused to TmrA. Incorporation efficiency of TmrA^EQ^B in proteoliposomes was analyzed by carbonate crush and sucrose density gradient floatation. Here, 95% of TmrA^EQ^B were reconstituted. These three parameters were used to calculate the average number of correctly-inserted TmrA^EQ^B complexes per proteoliposome. We added further information in the main text for clarification.

5) Figure 2B could benefit by having a darker contrast.

Thanks for the comment. We originally kept the illustration conservatively, however changed the panel accordingly.

6) The Results section is extremely condensed and hard to read. Since eLife does not have a firm word limit to manuscripts, the authors really should do better in explaining the (nice) experiments. In addition, I find the discussion a bit shallow, and disappointing considering the beautiful data.

Thanks for the comment. We have changed the main text accordingly.

7) Subsection “A single ATP turnover and phosphate release trigger the OF-to-IF return”: "as described above" Removal of ATP was not described above.

Thank you for your comment. We have corrected the main text.

8) Figure 2D and related supplement figure: The decay does not look monoexponential. I suspect that there are 2 different koff values: One for the dissociation of nucleotide from the canonical site (after hydrolysis) and the other for dissociation from the degenerate site. In that respect, it is surprising that in Figure 2F the nucleotide from the degenerate site does not seem to come off. Why does more than 80% of bound nucleotide dissociate in the experiment shown in 2D (this number indicates that dissociation from both sites takes place), but only 50% in 2F (this number suggests that ATP in the degenerate site does not come off).

Thank you for your comment. In scintillation proximity assays (SPA), ATP binding signals are slightly declining over time. This linear decrease is underlying the exponential decay of nucleotide dissociation. In principle, the signals could be corrected by the linear decrease. However, we decided to present the original raw data without further background subtraction. Regarding Figure2F, we analyzed the identity of nucleotides that were occluded by TmrA^EQ^B along the IF-to-OF switch after the removal of free ATP. After the hydrolysis event, hydrolysis products (ATP, ADP, and P_i_) are released from TmrA^EQ^B. Therefore, over time the ATP to ADP ratio is approaching 1/1. In the main text, we have clarified these differences.

9) Figure 2E: In the "No nucleotide" control there is a clear increase in bound peptide over the time course of the experiment. This must be explained. My guess is that ATP co-purifies with the protein.

Thank you for your comment. When measuring peptide binding by SPA, we observe a slight increase of binding over longer time periods. Since TmrA^EQ^B is immobilized on SPA beads, we suggest that peptide equilibration takes longer. We have changed the main text accordingly.

10) Figure 2—figure supplement 1 panel B: At 4 degrees, only 50% of bound nucleotide is released. Explain. It may be that the dissociation from the degenerate site is much slower at 4 degrees than at higher temp, or that the dissociation from the canonical site is much slower. To tell these two events apart would really give new insight.

Thank you for your comment. It is an interesting observation that the *k_off_* rate is decreased at 4°C while the amount of released nucleotide is reduced to 50%. We assume an asymmetric opening of both nucleotide-binding sites (NBSs) which is in line with previous reports (Barth et al., 2018, J Am Chem Soc). However, we can only speculate about the asymmetric contributions of both NBSs along the OF-to-IF transition.

11) Subsection “A single ATP turnover and phosphate release trigger the OF-to-IF return”: The difference in half-life at the different temperatures is very small. Given the large conformational change needed for the dissociation to occur, I find the marginal temperature dependence surprising. Explain.

Thank you for your comment. Since the OF-to-IF transition is initiated by a single turnover of ATP, we expected a large temperature dependency between 4°C, 20°C and 45°C. However, the physiological temperature of TmrAB is 68°C (Zutz et al., 2011), making it impossible to monitor nucleotide dissociation by SPA. For lower temperatures, the OF-to-IF return appears to show marginal temperature dependency.

12) Figure 1D and Figure 1—figure supplement 2 panel B: It is unclear whether these experiments are identical or not. If yes: why show it twice? If not: explain the difference better.

Thanks for your comment. We utilized two different substrate peptides and two distinct methods to confirm the ATP-induced abrogation of substrate binding. We have changed the main text accordingly.

13) Figure 1B and Figure 1—figure supplement 1 panel C: why are the absolute values (cpm) so different?

Thanks for your comment. In SPA, radiolabeled ligands are added in trace amounts to reduce background signals. In Figure1—figure supplement1C, the raw data is shown, while in Figure1B the signals are multiplied by the dilution factor of radiolabeled compound, resulting in much larger values. This is now explained more clearly in the figure legends.

14) Subsection “ATP binding drives substrate translocation monitored by single-liposome assays”: it is not clear what "untreated" refers to.

Thank you for your comment. We have changed the main text accordingly.

15) Figure 3—figure supplement 1 panel A. It is unclear what the panels show, and there is no explanation in the text. Why are the instant blue data different from the α-His data? The samples were incubated for the same amount of time, right? Why is there no quantification shown? From the qualitative images, the authors seem to draw quantitative conclusion. Please show proper quantification.

Thank you for your comment. The α-His immunoblot was used to determine the fraction of cleaved TmrA calculating the orientation of TmrAB in proteoliposomes. Equal loading of proteoliposomes was controlled by instant blue staining. We have explained this approach more clearly in the main text.

16) Figure 5C. I do not see biphasic behavior, and do not see the offset at 35 per liposome. I guess the wrong figure is shown?

Thanks for your comment. We observed an initial burst phase followed by a linear increase of the number of transported peptides per liposome for longer time periods. The offset of ~30 transported peptides per liposome resembles the y-intercept of the linear term of the regression function. We have explained this evaluation more clearly in the main text.

17) In general, the authors should not oversell their nice work by using words like "ultrasensitive", unscientific qualifications like "almost unlimited". Also, it does not help the reader if it is unclear what "this" and "these" refer to. For instance, in paragraph two of the Introduction "this question": there was no question asked. Paragraph three: “these questions”: again, I cannot find questions.

We have changed the main text accordingly.